# Antiaging Properties of *Kalanchoe blossfeldiana* Ethanol Extract—Ex Vivo and In Vitro Studies

**DOI:** 10.3390/molecules29235548

**Published:** 2024-11-24

**Authors:** Justyna Stefanowicz-Hajduk, Anna Nowak, Anna Hering, Łukasz Kucharski, Piotr Graczyk, Mariusz Kowalczyk, Tadeusz Sulikowski, Anna Muzykiewicz-Szymańska

**Affiliations:** 1Department of Biology and Pharmaceutical Botany, Medical University of Gdańsk, 80-416 Gdańsk, Poland; anna.hering@gumed.edu.pl (A.H.); petrilasendri21@gumed.edu.pl (P.G.); 2Department of Cosmetic and Pharmaceutical Chemistry, Pomeranian Medical University in Szczecin, 70-111 Szczecin, Poland; anna.nowak@pum.edu.pl (A.N.); lukasz.kucharski@pum.edu.pl (Ł.K.); anna.muzykiewicz@pum.edu.pl (A.M.-S.); 3Department of Biochemistry and Crop Quality, Institute of Soil Science and Plant Cultivation, State Research Institute, 24-100 Pulawy, Poland; mkowalczyk@iung.pulawy.pl; 4Clinic of General, Minimally Invasive and Gastroenterological Surgery, Pomeranian Medical University in Szczecin, 71-252 Szczecin, Poland; tadeusz.sulikowski@pum.edu.pl

**Keywords:** *Kalanchoe blossfeldiana*, hydrogels, antioxidant activity, skin penetration, elastase, hyaluronidase

## Abstract

Species of the genus *Kalanchoe* have a long history of therapeutic use in ethnomedicine, linked to their remarkable medical properties. These species include *Kalanchoe blossfeldiana* succulents, which grow in tropical regions. Despite the great interest in this plant, there are no reports about its therapeutic effects on the skin. In this study, the antioxidant properties of *K. blossfeldiana* ethanol extracts and the skin permeation of a topical hydrogel containing the extract (HKB) were assessed. Additionally, the content of active compounds in the *K. blossfeldiana* extract was evaluated by UHPLC-MS and HPLC-UV. The extract was analyzed with three antioxidant assays: ABTS, DPPH, and FRAP. Furthermore, the antielastase and antihialuronidase properties of the tested extract were assessed. Ex vivo penetration studies were performed using the Franz diffusion cells. The estimation of the cytotoxicity of HKB was performed by using an MTT assay ((4,5-dimethylthiazol-2-yl)-2,5-diphenyltetrazolium bromide) on the human fibroblasts HFF-1. The results obtained show that the antioxidant properties of *K. blossfeldiana* extract were similar to those of ascorbic acid, while antielastase and antihialuronidase tests indicated the strong antiaging and anti-inflammatory activity of the extract (IC_50_ was 26.8 ± 0.13 and 77.31 ± 2.44 µg/mL, respectively). Moreover, active ingredients contained in *K. blossfeldiana* extract penetrated through the human skin and accumulated in it. The cytotoxicity test showed that HKB had no significant effect on human fibroblasts at a concentration up to 0.5%. In conclusion, the hydrogel containing the *K. blossfeldiana* extract can be considered as an interesting and new alternative to dermatologic and cosmetic preparations.

## 1. Introduction

*Kalanchoe blossfeldiana* Poelln., belonging to the Crassulaceae family, grows naturally in tropical and subtropical regions and is commonly cultivated as a household and garden plant. The interest in this plant in recent years has been primarily due to its health benefits. The plant has long been used to treat different diseases. Many studies report its antioxidant, anticancer, and antibacterial effects [1,2,3,4]. In addition, *K. blossfeldiana* contains various secondary metabolites, including flavonoids such as kaempferol and quercetin as well as gallic and benzoic acid derivatives [1,3,5]. These metabolites showed, among other attributes, a high ability to scavenge free radicals and anti-inflammatory, antibacterial, and anticancer properties, which play a very important role in medicinal preparations, including those applied to the skin. This is important because, in recent years, more and more attempts have been made to use less popular plants and their extracts in preparations applied to the skin. There are no reports in the available literature on this type of *K. blossfeldiana* finding use. Also, extracts from *K. blossfeldiana* have never been investigated in terms of the penetration of the plant metabolites through the skin and their accumulation there.

The assessment of penetration and accumulation in the skin after the application of the preparation is very important because the vehicle used may significantly affect the release and penetration of active substances through the skin [6,7,8]. Moreover, in recent years, there has been a search for safe carriers that are easy to use and whose consistency allows for the easy placement of plant extracts in them [9]. Such vehicles include hydrogels, which are characterized by simple compositions and ease of use. Many authors have used hydrogels that contain various plant extracts in their studies. For example, hydrogel films composed of agarose, κ-carrageenan, and glycerol contained aqueous extracts of *Cryphaea heteromalla* with solid antioxidant activity [10]. Hydrogels containing 5% *Epilobium angustifolium* extract showed anti-inflammatory and antioxidant effects and also accelerated wound healing in vitro [11]. Moreover, with the addition of *Epilobium angustifolium* extracts, some phenolic acids contained in hydrogels penetrated the skin or accumulated in it, exhibiting additional antioxidant effects. In other studies, the hydrogels containing 0.5% of *Cannabis sativa* extracts had a moisturizing effect on the skin and affected the restoration of the hydrolipid balance and rebuilding of the hydrolipid barrier of skin damaged in the cleansing process by surfactants such as sodium laurosulfate [12].

The skin is the most external organ, with a complex structure, and for this reason it is capable of providing adequate protection to the body against the negative impacts of the environment. Macromolecules such as elastase, which gives the skin elasticity, or hyaluronic acid, which is responsible for the skin’s proper hydration and wound healing processes, are subject to enzymatic degradation, which increases with age. As a result, the skin loses its elasticity and, due to additional loss of hydration, becomes dry and less resistant to pathogen attacks. Wrinkles and inflammation, especially in the face, can be prevented by using cosmetics containing plant extracts, which have antioxidant effects and inhibit the action of enzymes such as elastase and hyaluronidase [13,14,15,16,17].

The aim of our study was to estimate antioxidant, antielastase, and antihialuronidase activity and assess the permeation of selected phenolic acids from a hydrogel with *K. blossfeldiana* ethanol extract (HKB) through the human skin and their accumulation in it. We also assessed the cytotoxicity of HKB on human fibroblasts HFF-1 in vitro.

## 2. Results

### 2.1. Phytochemical Profiles of K. blossfeldiana

The phytochemical analysis of the ethanol extract of *K. blossfeldiana* was performed with high-resolution mass spectrometry LC-QTOF-MS. Data analysis revealed the presence of at least 143 main components. Overall, 93 components were tentatively identified and classified into 23 arbitrarily established general compound categories (Table 1 and Appendix A). The most frequently occurring among the investigated extract were phenylpropanoid derivatives, observed mainly as glycosides (12 compounds), gallic acid derivatives and acyclic alcohol glycosides (10 and 9 compounds each, respectively), benzoic acid derivatives and organic acids (9 and 8 compounds each, respectively), and acyclic nitrile glycosides and flavanoles (7 and 6 compounds each, respectively). Among the benzoic acid derivatives group, it was possible to observe, among others, derivatives of vanillic and shikimic acid. The analyzed extract was rich in gallic acid derivatives, and these compounds were gallotannins. Subsequently, the following groups of compounds were identified in smaller quantities: megastigmane glycosides, flavonole glycosides, dimeric proanthocyanidins, phenol derivatives, and aminoacids (Table 1 and Appendix A).

Moreover, the content of selected phenolic acids in a hydrogel with *K. blossfeldiana* (HKB) was estimated and is presented in Table 2 and Figure 1. The following phenolic acids were found: gallic acid, protocatechuic acid, *p*-hydroxybenzoic acid, *m*-hydroxybenzoic acid, vanillic acid, gentistic acid, and hypogallic acid. The content of the analyzed phenolic acids ranges from 0.85 ± 0.11 µg·mL extract^−1^ for *m*-hydroxybenzoic acid to 284.74 ± 15.64 µg·mL extract^−1^ for gallic acid. The gallic acid and protocatechuic acid were the most abundant acids in the extract tested (Table 2).

### 2.2. Biological Activity of Ethanol Extract of K. blossfeldiana

#### 2.2.1. Antioxidant Activity

To estimate the antioxidant activity of ethanol extract of *K. blossfediana*, three antioxidant tests were performed. Ascorbic acid was used as a standard. The results obtained show that the extract had strong antioxidant properties, and in DPPH and ABTS assays the IC_50_ values for the extract were lower than those obtained for ascorbic acid (Table 3). The highest activity of the extract was observed in the ABTS test. In the FRAP test, the IC_50_ value of *K. blossfeldiana* extract was about two times higher than that of the standard compound. All the tests confirmed the potent antiradical and reduction activity of the plant extract.

#### 2.2.2. Enzyme Inhibition

In this study, the activity of the *K. blossfeldiana* ethanol extract on skin enzymes was estimated using elastase and hyaluronidase. The results obtained indicate that the tested extract inhibited the activity of both enzymes, and this effect was dose-dependent. In the case of hyaluronidase, the extract caused the complete inhibition of the enzyme at a concentration of 400 µg/mL (Figure 2), while elastase was completely inhibited at an extract concentration of 50 µg/mL (Figure 3). The IC_50_ values calculated for the extract were only 1.5 times higher in comparison to the standard—oleanolic acid—used in the experiments (Table 4).

### 2.3. Stability Test of HKB

The tested hydrogel with the *K. blossfeldiana* ethanol extract showed appropriate physical properties. After performing the vortex test, no separation of the extract was observed. Similarly, we did not observe changes in the color and odor of HKB in comparison to HKB before the heating–cooling test.

### 2.4. Ex Vivo Study with HKB

#### 2.4.1. Permeation Through the Skin

The results of the permeation of phenolic acids from HKB during the 24 h study are shown in Table 5. The highest degree of penetration was observed for gallic acid and protocatechuic acid, the cumulative masses of which, when collected after 24 h of permeation, were 249.73 ± 13.69 and 97.55 ± 5.31 µg·cm^−2^, respectively. It was observed that these two acids penetrated through human skin the fastest. Gallic acid penetrates the skin within the first hour of applying hydrogel to the skin, while protocatechuic acid was identified in the acceptor fluid collected after the second hour of the study. The remaining acids penetrated the skin much slower and were only identified in the acceptor fluid in the last hours of the experiment.

#### 2.4.2. Accumulation in the Skin

The accumulation of phenolic acids in the skin 24 h after the application of the hydrogel to the skin is presented in Figure 4. All analyzed phenolic acids accumulated in the skin, except for hypogallic acid. It was found that gallic acid (107.58 ± 15.27 µg·g skin^−1^) and protocatechuic acid (72.28 ± 8.08 µg·g skin^−1^) accumulated the most in the skin.

### 2.5. In Vitro Study with HKB

To estimate the viability of HFF-1 cells after treatment with 0.5%, 1%, and 5% HKB, we added the hydrogels to the wells with fibroblasts. The results obtained indicate that the highest viability of the cells was observed in the experiments with 0.5% HKB (Figure 5C). The percentages of viable cells were 94.99 ± 7.77, 89.07 ± 8.74, 68.42 ± 0.31, 65.35 ± 5.89, and 69.79 ± 7.08% for the calculated extract amounts (8, 16, 32, 64, and 320 µg/cm^2^, respectively). There were similar results for 1% HKB, with cell viability results of 89.98 ± 7.17, 69.53 ± 8.79, 69.12 ± 9.37, 66.45 ± 8.55, and 45.64 ± 3.26% for the calculated extract amounts of 16, 32, 64, 320, and 640 µg/cm^2^, respectively (Figure 5B). We observed the lowest viability for 5% HKB, with values of 62.97 ± 5.98, 54.26 ± 4.24, 40.38 ± 4.02, 32.26 ± 0.78, and 35.02 ± 1.52% for the calculated extract amounts of 32, 80, 640, 960, and 1600 µg/cm^2^, respectively (Figure 5A). The results obtained with the MTT assay were confirmed via microscopic observation (Figure 6).

This part of the study indicates that hydrogels with *K. blossfeldiana* extracts may have potential uses on the skin at a concentration up to 0.5%. However, it should be emphasized that these results were obtained using isolated fibroblasts and that further study on living skin should be performed to confirm the safety of hydrogels with *K. blossfeldiana* extracts.

## 3. Discussion

Plants are a valuable source of bioactive compounds and are important for the pharmaceutical and cosmetic industries. In recent years, interest in the use of plant extracts for the production of dermatological and cosmetic preparations has increased significantly. The main reason for this is that valuable secondary metabolites are found in large amounts in plants, which have primarily antioxidant, but also anti-inflammatory, antibacterial, and antiaging effects [18,19]. In recent years, there has been growing interest in lesser-known plants that could have high potential in relation to health. *Kalanchoe blossfeldiana*, which is still little known in this respect, has attracted increasing interest. There are not many reports in the available literature on the identification of individual phenolic acids in *K. blossfeldiana* leaves. Previous analysis of aqueous fractions from the extract of *K. blossfeldiana,* performed using high-resolution mass spectrometry data, showed the presence of benzoic acid derivatives as well as a high content of gallic acid derivatives [3]. In our study, phytochemical analysis revealed that the most frequently detected groups of compounds in the *K. blossfeldiana* ethanol extract were phenylpropanoid derivatives. In this group, the metabolites identified were coumaroyl, feruloyl, and caffeoyl derivatives, which were previously identified in *K. blossfeldiana* and *K. pinnata* [3,20,21]. Other important groups identified in the extract we analyzed were derivatives of benzoic acid and gallic acid. This was additionally confirmed by HPLC-UV analysis. It contained gallic acid, protocatechuic acid, *p*-hydroxybenzoic acid, *m*-hydroxybenzoic acid, vanillic acid, gentistic acid, and hypogallic acid. In our study, gallic acid was detected in the highest amounts. A similarly high content of this compound in the leaves of *K. blossfeldiana* was confirmed by Pryce [22]. Phenolic acids are important plant components that have an impact on skin health. The main property of these compounds is high antioxidant activity. When applied to the skin, they protect it from the excessive oxidative stress caused by reactive oxygen species (ROS). Excessive ROS production leads to many negative effects, including cell damage [23]. The natural antioxidants contained in plants play a key role here as they support the endogenous skin defense system. The *K. blossfeldiana* ethanol extract applied to the skin in our study was characterized by high antioxidant activity. High levels of ROS may hinder faster wound healing, especially chronic changes. Providing exogenous antioxidants stimulates cell migration and angiogenesis. Therefore, the addition of natural antioxidants to cosmetics/dermatologic preparation may be very effective in this case. For example, applying hydrogel with antioxidant activity to the skin can reduce oxidative stress, improve the wound microenvironment, and ultimately achieve rapid skin repair [24]. Moreover, to keep the skin in good condition for as long as possible, it is necessary to use preparations with ingredients that inhibit the activity of skin enzymes—components of the extracellular matrix (ECM). The *K. blossfeldiana* extract can inhibit elastase and hyaluronidase significantly and this effect is slightly weaker than the effect of oleanolic acid. 

In the case of extracts containing many different chemical compounds, studies have not been conducted in relation to the mechanisms of enzymatic reactions. Extracts are considered to be mixed mechanisms, because the molecules present in the extract can affect both the enzyme itself and the enzyme–substrate complex. In the mixture of compounds, it is impossible to estimate which of the compounds has a stronger inhibitory effect on enzyme activity: the one present in the largest quantity or the one with a higher affinity to the active site but a lower concentration. In our study, gallic acid had the highest concentration among the analyzed compounds. Plant extracts with this acid as one of the main components are effective inhibitors of both elastase and hyaluronidase [25,26]. 

Antioxidants contained in plant extracts applied to the skin can accumulate in it or penetrate into its deeper layers. Therefore, when designing new cosmetic/dermatological preparations, the study of skin penetration plays an increasingly important role. Active substances are first released from the preparation and then penetrate the stratum corneum (SC). In addition, penetration can be limited by factors such as lipophilicity or the chemical structure of the active substance itself [27,28,29]. The SC is a thin membrane consisting primarily of cornified epidermal cells, while the main components are lipids, namely cholesterol, ceramides and its esters, and fatty acids [30,31]. The analysis of the penetration of phenolic acids through the skin as well as the assessment of their accumulation therein is an important step in the selection of an appropriate carrier for plant extracts. Plant extracts contain a whole pool of secondary metabolites with different physicochemical properties. In our research, we assessed the penetration of selected phenolic acids from a hydrogel containing dry ethanol extract from leaves of *K. blossfeldiana*. The penetration test was performed in a Franz diffusion cell using the human skin. There are no reports in the available literature about the application of *K. blossfeldiana* extracts to the skin and the subsequent analysis of the penetration and accumulation of phenolic acids. Therefore, in the next stage of our research, the penetration of phenolic acids from the hydrogel containing extract of *K. blossfeldiana* was assessed. After the application of HKB on the skin, the phenolic acids penetrated the skin to varying degrees. Some phenolic acids, such as gallic acid and protocatechuic acid, penetrated the skin in the first and second hour after application. The highest permeation values were observed for these phenolic acids. Some of the phenolic acids analyzed in our study were only identified in the acceptor fluid after the 8th hour of the experiment and these were *p*-hydroxybenzoic acid, *m*-hydroxybenzoic acid, and vanillic acid. However, gentistic acid only penetrated the skin after 24 hours, and in the case of hypogallic acid, the 24 h test period was too short for this compound to cross the skin barrier. As we know, plant extracts contain various secondary metabolites which, depending on their concentrations, may interact synergistically, which may also increase the permeation of some of them. The concentration of gallic acid and protocatechuic acid was the highest in the prepared plant extract, which was probably reflected in the fact the highest penetration was seen for these compounds. Previous authors also reported the rapid skin penetration of gallic acid from microemulsions containing *Glochidion wallichianum* extract and *E. angustifolium* ethanolic extract [32,33]. However, some authors have shown very low or negligible penetration of phenolic acids from preparations containing plant extracts. For example, phenolic acids were released to a very small extent from a hydrogel containing 5% coffee seed extract [28]. Conversely, the rosmarinic acid contained in *Plectranthus ecklonii* did not permeate the human skin, neither from the water solution nor from the ethanol/PG solution prepared from this plant [34]. Of course, the vehicle used plays a key role in releasing active substances into the skin. In our study, we chose hydrogel as the vehicle for preparation. The simple composition of hydrogels, as well as their high water contents and high biocompatibility with skin cells, makes them very popular vehicles. Hydrogels are playing an increasingly important role in pharmacy and cosmetology and they seem to be one of the most promising groups of biomaterials [9]. Hydrogels containing plant extracts with many active substances can have a wide range of effects, including antibacterial, anti-inflammatory, and antioxidant effects [9,35,36]. For example, phenolic acids contained in *E. angustifolium* extracts penetrate the skin from the hydrogel better than they do from the emulsion [37]. Conversely, Žilius et al. showed the highest degree of penetration of phenolic acids from a hydrogel containing a propolis extract, which rapidly released acids such as coumaric, caffeic, and ferulic acids. These authors suggested that the higher viscosity of the emulsions containing oil content hampers the diffusion of these substances [38]. 

Active substances must penetrate the stratum corneum to reach the cells localized in the lower strata of the epidermis and the dermis. Many authors suggest that, in the case of cosmetic preparations including plant extracts, greater accumulation in the skin is preferred. This is because extracts show, among other impacts, antioxidant or antiaging effects [11,28,39]. In our study, most of the analyzed phenolic acids accumulated in the skin, with gallic acid and protocatechuic acid being the most abundant. Taking into account the beneficial effects of cosmetic or dermatological preparations on the skin, the accumulation of some of the analyzed compounds in the skin may be very important. For example, a gallic acid can exhibit a variety of biological activities with regard to the skin including antioxidant [27,40,41], anticancer [42,43,44], anti-inflammatory [43], antibacterial [45,46], and antiaging effects [47]. 

## 4. Materials and Methods

### 4.1. Chemicals

Ethanol, methanol, acetic acid, and acetone were obtained from Chempur (Piekary Śląskie, Poland). Propylene glycol and hydroxyethylcellulose were obtained from Pol-Aura (Morąg, Poland), and acetonitrile for HPLC was obtained from J.T. Baker (Berlin, Germany). Hypogallic acid, protocatechuic acid, gentistic acid, and vanillic acid were obtained from Sigma Aldrich (Steinheim am Albuch, Germany); *p*-hydroxybenzoic acid, *m*-hydroxybenzoic acid, gallic acid, and phosphate-buffered saline (PBS; pH 7.00 ± 0.05) were obtained from Merck (Darmstadt, Germany). Furthermore, the 31 reference standards (listed in [48]) were obtained from Merck (Darmstadt, Germany) and used for the calibration of the semi-quantitation method described previously [3]. In antioxidant assays and tests with hyaluronidase and elastase, DPPH (2,2-diphenyl-1-picrylhydrazyl), ABTS (diammonium 2,2′-azinobis[3-ethyl-2,3-dihydrobenzothiazole-6-sulphonate]), TPTZ (2,4,6-Tris(2-pyridyl)-s-triazine), potassium persulfate, neutrophil elastase, hyaluronidase (from bovine testes, 400–1000 U/mg), hyaluronic acid, bovine serum albumin (BSA), N-Sccinyl-Ala-Ala-Ala-p-nitroanilide (SANA), ascorbic acid, and oleanolic acid were obtained from Merck Millipore (Burlington, MA, USA). TRIS-HCl, HCl (77 mM), acetate buffer (0.3 M, pH 3.6), sodium acetate, sodium phosphate, phosphoric acid, and FeCl_3_ × 6 H_2_O were purchased from P.O.Ch. (Gliwice, Poland). 

### 4.2. Plant Material and Extraction

The pink-flowered cultivar of *K. blossfeldiana* was obtained from a commercial garden (Garden Center Justyna, Gdansk, Poland) and one specimen was deposited in GDMA Herbarium (Herbarium of the Medical University of Gdansk, No. 21759). The plant leaves were divided into two parts and used for the preparation of the extracts. Initially, the leaves (100 g) were macerated and stirred with 95% ethanol (0.5 L) for 24 h at RT. The ethanol extract was filtered, concentrated under reduced pressure at 40 °C, and lyophilized. The lyophilizates were dissolved in DMSO and submitted for antioxidant, antielastase, and antihialuronidase analyses.

The extract needed to produce the hydrogel was prepared from the second part of the leaves. Namely, after drying at RT in a well-ventilated area to a constant weight, 2.5 g of dried leaves was extracted with 100 mL 70% (*v*/*v*) ethanol for 60 min in an ultrasonic bath (Polsonic Sonic 5, POLSONIC Palczynski, Warsaw, Poland) at a frequency of 40 kHz. After filtration, the extract was divided into two parts. The first part was transferred for HPLC analysis and the second part was evaporated under reduced pressure at 40 °C and used to prepare a hydrogel. All the samples were stored in the dark at 4 °C until further analysis and hydrogel preparation. 

### 4.3. Phytochemical Analysis of K. blossfeldiana Ethanol Extract

Semi-quantitative data were acquired following the methodology outlined in a prior publication [3]. For UHPLC-MS analyses, samples were prepared utilizing a simplified protocol adapted from Salem et al. [49]. Specifically, 10 mg (±0.1 mg) of dried extract underwent a 45 min incubation at 4 °C with a mixture of methyl-tertbutyl ether and methanol (in a 3:1 *v*/*v* ratio). Subsequently, the samples were sonicated in an ice water bath for 15 min before adding a distilled water/methanol mixture (in a 3:1 *v*/*v* ratio). Following centrifugation, the water/methanol phases were collected, evaporated under nitrogen (at 30 °C), and reconstituted in methanol (1 mL of 40%). Three replicate samples were prepared, each containing 10 mg of the original extract per mL. 

Prior to analysis, the reconstituted samples underwent filtration through 0.2 µm cellulose centrifugal filters. The UHPLC-MS analyses were conducted using a Thermo Ultimate 3000 RS-coupled Bruker Impact II QTOF mass spectrometer (Bruker, Billerica, MA, USA). Separations were performed on a Waters HSS T3 column with a constant flow rate of 0.4 mL/min using 0.1% (*v*/*v*) formic acid (mobile phase A) and acetonitrile with 0.1% (*v*/*v*) formic acid (mobile phase B). The elution profile began with 5% of phase B for 1 min, followed by linear gradients of phase B, rising from 5% to 14% over 7.5 min and from 14 to 48% over 18.5 min. The concentration of the mobile phase was then increased over 4.5 min to 95% of phase B, maintained at that level for 7 min, and then equilibrated for 5 min with 5% of phase B. The column effluent was proportioned 3:1, with a fixed flow splitter placed between the charged aerosol detector (CAD, Thermo Corona Veo RS, Thermo Fisher Scientific, Waltham, MA, USA) and the ion source of the QTOF. Ions were measured in the negative electrospray mode in the *m*/*z* range of 100–1200 at a 5 Hz scanning frequency. MS/MS spectra were obtained in DDA (data-dependent analysis) mode, with the two most intense precursors in each scan fragmented by CID (collision-induced dissociation, Ar collision gas). 

The concentration of each analyte could not be quantified in an absolute manner due to the lack of commercially available reference standards. Instead, observed analytes were semi-quantified using signals from charged aerosol detectors. The influence of the mobile phase on the signal intensity was investigated using 31 reference standards differing in retention time. The concentration of each analyte was then calculated as a weighted average of the responses of the pair of adjacent calibration standards, employing RT lower and higher than the RT of the analyte. After each block of 10 regular sample injections, a pooled quality control sample was analyzed. Data processing and analysis were performed using Bruker Data Analysis (ver. 4.4SR1). High-resolution *m*/*z* measurements were used for the preliminary, tentative identification of metabolites. Chemical formulas were calculated based on these results. The obtained formulas were validated based on the acquired MS/MS spectra.

The high-performance liquid chromatography HPLC-UV method (Knauer, Berlin, Germany) was used to assess selected phenolic acids in the tested plant extract. The mobile phase contained 1% acetic acid and MeOH in a ratio of 92:8, *v*/*v*. A 125 mm × 4 mm C18 column containing Eurospher 100 with a particle size of 5 μm was used at a flow rate of 1 mL/min. The amount of the tested extract injected into the column was 20 μL. Individual phenolic acids were identified based on patterns and retention times. The results are presented as mean ± standard deviation (±SD). All samples were analyzed in triplicate. The following standards were used: gallic acid (r = 0.9999, y = 30106x − 1.2008, tR—4.944 min); protocatechuic acid (r = 0.9998, y = 21474x − 1.3075, tR—9.123 min); *p*-hydroxybenzoic acid (r = 0.9929, y = 17279x + 3.623, tR—26.892 min); *m*-hydroxybenzoic acid (r = 0.9997; y = 65401x − 0.0413, tR—18.776 min); vanillic acid (r = 0.9998, y = 17122x − 0.2184, tR—19.727 min); gentistic acid (r = 0.9995, y = 14706x − 0.8531, tR—16.394 min); and hypogallic acid (r = 0.9999; y = 51231x + 2.0769, tR—24.583 min). 

### 4.4. Antioxidant Tests

#### 4.4.1. DPPH Assay

The DPPH method [3,50] was used for the assessment of the antioxidant activity of the ethanol extract of *K. blossfeldiana*. Briefly, the ethanol extract (dissolved in DMSO) was mixed with DPPH (0.06 mM) and its absorbance was measured by a 96-well microplate reader at λ = 510 nm (Epoch, BioTek System, Winooski, VT, USA). The standard was ascorbic acid, and the control was a DPPH solution with DMSO.

DPPH inhibition was calculated as follows:DPPH Inhibition (%) = [(Control − Sample)/Control] × 100%

The program GraphPad Prism v. 9.0.0 (GraphPad Software, Boston, MA, USA) was used for calculation of IC_50_ values for the extract and ascorbic acid.

#### 4.4.2. ABTS Assay

The ABTS method was used for the estimation of the antioxidant activity of the ethanol extract of *K. blossfeldiana* [3,51]. Briefly, the ethanol extract (dissolved in DMSO) was mixed with ABTS solution (3.5 mM potassium persulfate, 2 mM ABTS diammonium salt). The absorbance of the solution was analyzed with a 96-well microplate reader at λ = 750 nm (Epoch, BioTek System, Winooski, VT, USA). The standard was ascorbic acid, and the control was the ABTS solution with DMSO.

ABTS inhibition was calculated as follows:ABTS Inhibition (%) = [(Control − Sample)/Control] × 100%

The program GraphPad Prism v. 9.0.0 (GraphPad Software, Boston, MA, USA) was used for the calculation of IC_50_ values for the extract and ascorbic acid.

#### 4.4.3. FRAP Assay

The FRAP assay was used for the estimation of the reducing ability of the ethanol extract of *K. blossfeldiana*. Briefly, the extract was mixed with FRAP reaction mixture [52]. The absorbance was read with a 96-well microplate reader at λ = 593 nm (Epoch, BioTek System, Winooski, VT, USA). The calibration curve, plotted for ascorbic acid as a standard (1–1000 µg/mL), was used for the determination of the percentage of reduced iron ions.

The program GraphPad Prism v. 9.0.0 (GraphPad Software, Boston, MA, USA) was used for the calculation of the IC_50_ values of the extract and ascorbic acid.

### 4.5. Inhibition of Enzymes

#### 4.5.1. Antihyaluronidase Assay

The inhibition of hyaluronidase activity by the ethanol extract of *K. blossfeldiana* was evaluated spectrophotometrically [53]. We incubated sodium phosphate buffer hyaluronidase and a series of dilutions of the analyzed extract or standard (37 °C, 10 min). After mixing with hyaluronic acid (HA), the mixture was then incubated (37 °C, 45 min). An acid albumin solution was used for the precipitation of the undigested HA. A 96-well microplate reader (Epoch, BioTek System, Winooski, VT, USA) was used to measure the absorbance of the reaction mixture (λ = 600 nm). The standard was oleanolic acid.

The percentage of inhibition was calculated as follows:Hyaluronidase inhibition = [(Aextract − Acontrol)/Ahyaluronic acid − Acontrol] × 100%

The program called GraphPad Prism v. 9.0.0 (GraphPad Software, Boston, MA, USA) was used for the calculation of IC_50_ values for the extract and oleanolic acid.

#### 4.5.2. Antielastase Assay

The activity of the extract on enzyme elastase was assessed according to the previously described method [54]. Briefly, the ethanol extract was mixed with porcine pancreatic elastase and Tris-HCl buffer (pH 8.0). After incubation, the substrate—SANA—was added and absorbance was measured with a plate reader (Epoch, BioTek System, Winooski, VT, USA) with respect to *p*-nitroaniline at λ = 410 nm every 20 s for 20 min. The standard was oleanolic acid.

The percentage of inhibition was calculated as follows:Elastase inhibition = [(Acontrol − Aextract)/Acontrol] × 100%

The program called GraphPad Prism v. 9.0.0 (GraphPad Software, Boston, MA, USA) was used for the calculation of IC_50_ values for the extract and oleanolic acid.

### 4.6. Prepared HKB Hydrogel

The hydrogel was prepared according to a modified version of the procedure outlined by Zagórska-Dziok et al. [12]. Overall, 0.25 g of the evaporated extract was dissolved in 0.5 g of propylene glycol and then suspended in 4.25 g of gel containing 1% hydroxyethylcellulose (HEC). HEC was added to distilled water and mixed on a magnetic stirrer (Chemland MS-H280-Pro, Chemland, Stargard, Poland) at 40 °C and 250 rpm. The final concentration of the dry extract in the hydrogel was 5% (HKB).

### 4.7. Stability of Hydrogel with K. blossfeldiana Extract

The stability of the hydrogel was tested based on the method previously described by Muthachan and Tewtrakul [55]. The stability of the hydrogel was assessed via a heating–cooling test: this involved incubation at 45 °C (drying oven, DHG-9075A) for 48 h, followed by incubation at 4 °C (refrigerator) for 48 h. The stability test was repeated six times. During the stability test, the preparation was also visually assessed. The phase separation test of the preparation was evaluated using a centrifuge test. Samples of the preparation, measured in 1 g amounts, were centrifuged (MPW-223e, Mechanika Precyzyjna, Warsaw, Poland) at 4000× *g* rpm at 25 °C for 10 min.

### 4.8. Ex Vivo Skin Permeation Studies

#### 4.8.1. Human Skin

This study used human skin obtained from the abdomen of living patients as a result of plastic surgery. The material was collected with the consent of the patients, and the study was approved by the Ethical Committee of the Pomeranian Medical University in Szczecin, No. KB0012/02/18. The obtained fresh skin was washed several times in PBS buffer (pH 7.4). Then, the skin was cut into 0.5 mm thick pieces using a dermatome and divided into pieces measuring 2 cm × 2 cm. The skin samples were stored at −20 °C and wrapped in aluminum foil. The material was stored for no longer than 3 months. This storage time does not change the barrier properties of the skin [56]. 

#### 4.8.2. Permeation Studies

The permeation study was performed using Franz diffusion cells (SES GmbH Analyse Systeme, Bechenheim, Germany) with diffusion areas of 1 cm^2^. The diffusion cells were kept at a constant temperature of 37.0 ± 0.5 °C [57], which was maintained via a thermostat (VEB MLW Prüfgeräte-Werk type 3280, Leipzig, Germany). The volume of the acceptor chamber was 8 mL, and the volume of the donor chamber was 2 mL. The contents of the acceptor chamber were constantly mixed using a magnetic stirrer. After placing the skin between the donor and acceptor chambers, its integrity was tested using an LCR 4080 meter (Voltcraft LCR 4080, Conrad Electronic, Hirschau, Germany) [58]. The penetration test was conducted for 24 h. Then, 0.3 mL of acceptor fluid was collected after 1, 2, 3, 5, 8, and 24 h, after which the chamber was refilled with the same volume of PBS (pH 7.4). The analyzed hydrogel in the amount of 1 g was applied to the donor chamber. The collected acceptor fluid was sent for HPLC analysis to determine the content of the selected phenolic acids. This was calculated as cumulative mass (µg). 

### 4.9. Accumulation of the Phenolic Acids in the Skin

The assessment of phenolic acid accumulation in the skin after the completion of 24 h of permeation was performed just as in our previous studies [32]. After the completion of the permeation study, the skin was collected and rinsed several times with 0.5% sodium lauryl sulfate solution. Then, the diffusion area was weighed and cut into small pieces, which were incubated in 2 mL of methanol for 24 h. After incubation, the skin samples were homogenized using a homogenizer (IKA ^®^T18 digital ULTRA TURRAX, Staufen, Germany). The supernatant was centrifuged and analyzed with HPLC to assess the phenolic acid content. The accumulation of phenolic acids in the skin was expressed as the mass of phenolic acid per mass of skin (µg·g skin^−1^). 

### 4.10. In Vitro Study with HKB

#### 4.10.1. Human Fibroblasts

Human fibroblasts HFF-1 was purchased from the LGC Standards (Teddington, Mid-dlesex, UK) and cultured in a DMEM medium with high levels of glucose and streptomycin (100 mg/mL), penicillin (100 units/mL), and 10% (*v*/*v*) fetal bovine serum (FBS). The fibroblasts were incubated at 37 °C and 5% CO_2_. 

#### 4.10.2. Viability Test

To estimate the percentage of viable cells after treatment with HKB, we seeded HFF-1 cells in 96-well plates (5 × 10^3^ cells/well) and added 0.5%, 1%, and 5% HKB at a volume of 0.1–20 µL/well. The amount of the *K. blossfeldiana* extract in each well ranged from 2.5 to 500 µg, depending on the HKB concentration. The area of the plate wells was 0.32 cm^2^. After 24 h, the cells were treated with MTT ((4,5-dimethylthiazol-2-yl)-2,5-diphenyltetrazolium bromide). The formazan crystals obtained were dissolved in DMSO and the absorbance was measured at 570 nm (with the plate reader Epoch, BioTek Instruments, Winooski, VT, USA). The viability of the cells was compared to the control, i.e., HFF-1 cells treated with hydrogel (20 µL) without the extract. 

### 4.11. Statistical Analysis

Results are presented as the mean ± standard deviation (±SD). The Student’s *t*-test was used to compare the results with those of the standard compounds or control. The statistical significance was set at *p* < 0.05. 

## 5. Conclusions

*Kalanchoe blossfeldiana* ethanol extract may be a valuable component of dermatological and cosmetic preparations due to its high antioxidant, antihyaluronidase, and antielastase activity. The phenolic acids contained in the plant extract accumulate in the human skin and may play a crucial role in the protection of this barrier against damaging factors. These results will be the basis for further research on living organisms in the future.

## Figures and Tables

**Figure 1 molecules-29-05548-f001:**
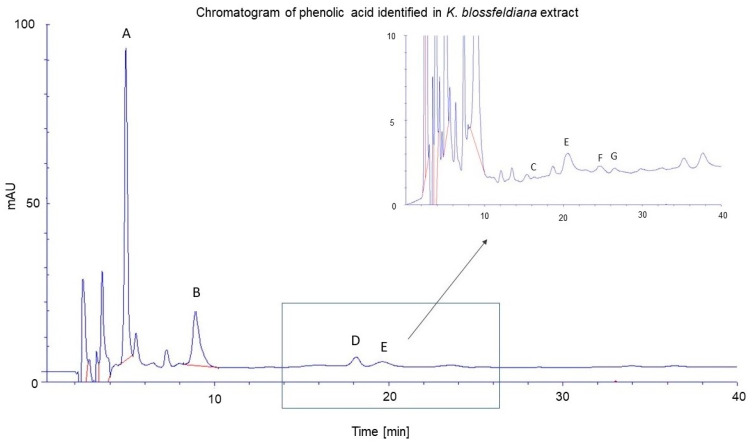
Chromatogram of phenolic acid identified in *K. blossfeldiana* ethanol extract from leaves. A—gallic acid; B—protocatechuic acid; C—gentistic acid; D—*m*-hydroxybenzoic acid; E—vanilic acid; F—hypogallic acid; G—*p*-hydroxybenzoic acid. The samples were diluted tenfold before HPLC analysis.

**Figure 2 molecules-29-05548-f002:**
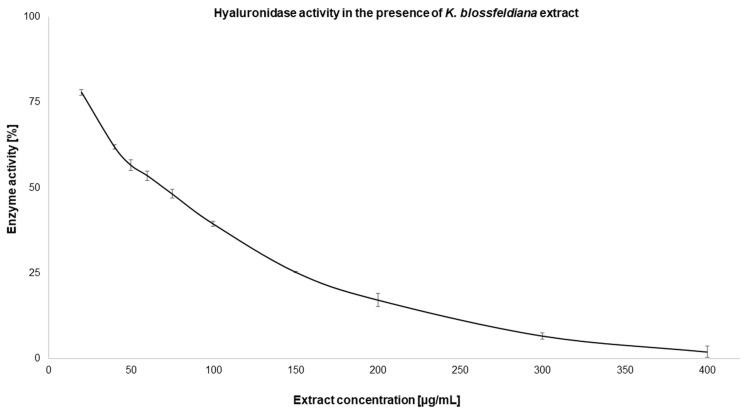
Hyaluronidase activity in the presence of a *K. blossfeldiana* ethanol extract. The experiment was performed in three independent repetitions (*n* = 9). Error bars represent standard deviations.

**Figure 3 molecules-29-05548-f003:**
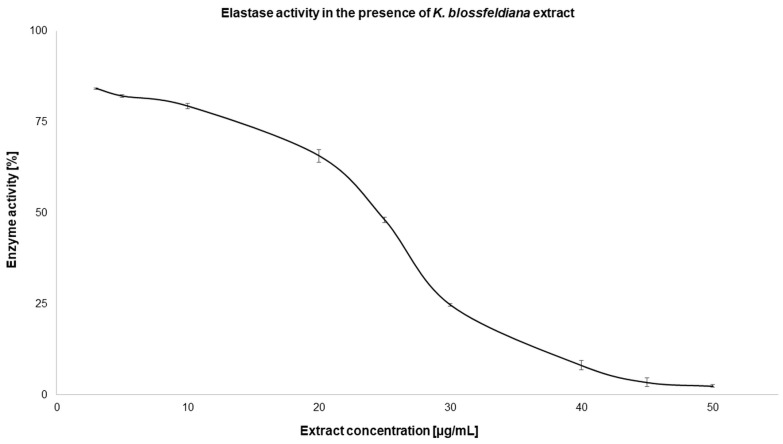
Elastase activity in the presence of a *K. blossfeldiana* ethanol extract. The experiment was performed in three independent repetitions (*n* = 9). Error bars represent standard deviations.

**Figure 4 molecules-29-05548-f004:**
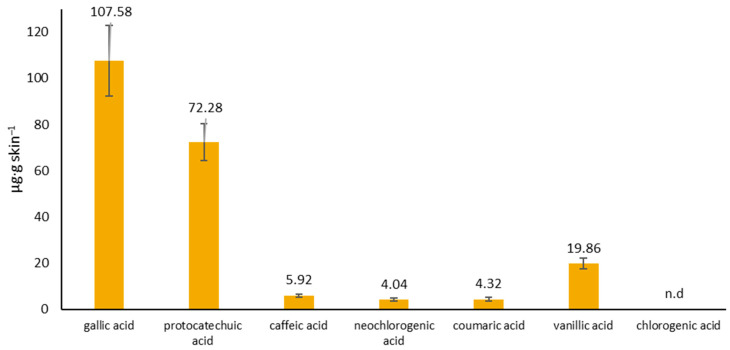
The accumulation of phenolic acids in the skin. The content of individual phenolic acids was determined in the skin extraction fluid collected after the 24 h penetration study. All values are presented as mean ± SD, where n = 3; n.d.—not detected.

**Figure 5 molecules-29-05548-f005:**
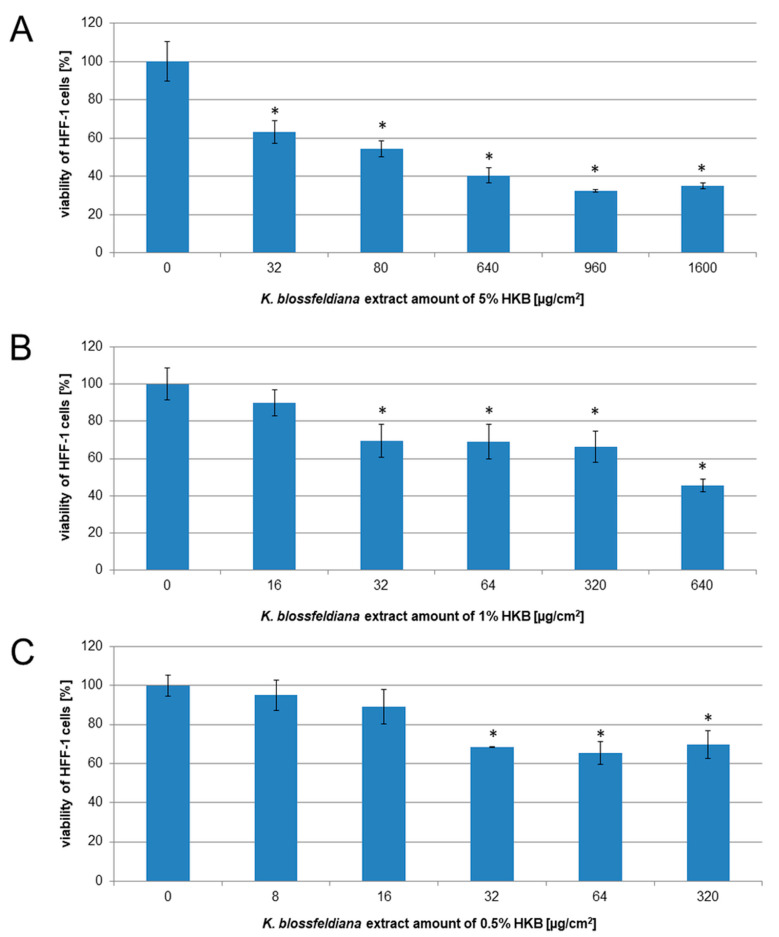
The viability of HFF-1 cells treated with HKB at concentrations of 5% (**A**), 1% (**B**), and 0.5% (**C**) for 24 h. The results were obtained via an MTT assay and they are presented as the mean values of two experiments performed in three repetitions. Error bars indicate the standard deviation (±SD). Asterisks indicate significant differences (Student’s *t*-test, *p* < 0.05) in comparison to the control (the cells treated with hydrogel without the extract).

**Figure 6 molecules-29-05548-f006:**
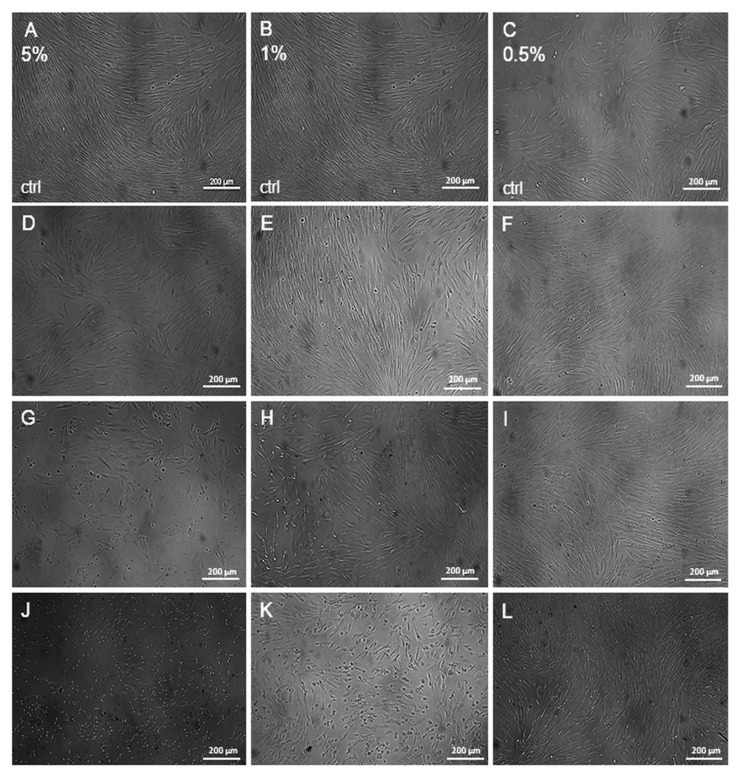
The microscopic observation of the viability of HFF-1 cells treated with HKB at concentrations of 5%, 1%, and 0.5% for 24 h. (**A**–**C**)—the control treated with hydrogel without the extract; (**D**,**G**,**J**)—cells treated with the extract amounts of 32, 640, and 1600 µg/cm^2^ of 5% HKB; (**E**,**H**,**K**)—cells treated with the extract amounts of 32, 320, and 640 µg/cm^2^ of 1% HKB; (**F**,**I**,**L**)—cells treated with extract amounts of 16, 64, and 320 µg/cm^2^ of 0.5% HKB; magnification ×100.

**Table 1 molecules-29-05548-t001:** Categories of compound detected in the ethanol extract of *K. blossfeldiana* by LC-QTOF-MS.

	Compounds Categories	Total Number of Compounds in Each Group
1	Carbohydrate	1
2	Organic acid	8
3	Acyclic alcohol glycoside	9
4	Acyclic nitrile glycoside	7
5	Gallic acid derivative	10
6	Aminoacid	3
7	Acyclic acid glycoside	2
8	Benzoic acid derivative	9
9	Acetophenone derivative	1
10	Phenylpropanoid derivative	12
11	Phenol derivative	3
12	Flavanole	6
13	Sesquiterpenoid derivative	1
14	Dimeric proanthocyanidin	5
15	Megastigmane glycoside	3
16	Dimeric iridoid derivative	1
17	Monoterpene derivative	1
18	Bicyclo [3.1.1] glycoside	1
19	Phenylethane glycoside	1
20	Flavonole glycoside	5
21	Fatty acid glycoside	1
22	Iridoid glycoside	1
23	Lipid	2
24	Unidentified	50
Total		93

**Table 2 molecules-29-05548-t002:** Phenolic acid content in HKB.

Phenolic Acid	µg·mL Extract^−1^
Gallic acid	284.74 ± 15.64
Protocatechuic acid	74.35 ± 4.30
*p*-hydroxybenzoic acid	11.10 ± 0.43
*m*-hydroxybenzoic acid	0.85 ± 0.11
Vanillic acid	12.63 ± 0.33
Gentistic acid	9.10 ± 0.39
Hypogallic acid	1.95 ± 0.07

The results obtained from the three experiments are presented as mean values ± standard deviation (±SD).

**Table 3 molecules-29-05548-t003:** The antioxidant activity of the *K. blossfeldiana* ethanol extract as determined with DPPH, ABTS, and FRAP tests.

Test	IC_50_ [µg/mL]	
	*K. blossfeldiana* Ethanol Extract	Ascorbic Acid
DPPH	7.72 ± 0.09 *	15.23 ± 0.76
ABTS	4.21 ± 0.32 *	7.38 ± 0.09
FRAP	11.25 ± 0.17 *	5.29 ± 0.21

The values were obtained from three experiments performed three times (*n* = 9). Significant differences relative to ascorbic acid (as a standard) are marked with an asterisk “*” (*p* < 0.05).

**Table 4 molecules-29-05548-t004:** The activity of the *K. blossfeldiana* ethanol extract against hyaluronidase and elastase.

Enzymatic Inhibition Assay	IC_50_ [µg/mL]	
	*K. blossfeldiana* Ethanol Extract	Oleanolic Acid
Hyaluronidase	77.31 ± 2.44 *	49.33 ± 1.35
Elastase	26.8 ± 0.13 *	17.25 ± 0.27

The values were obtained from three experiments performed in three repetitions (*n* = 9). Significant differences relative to oleanolic acid (as a standard) are marked with an asterisk “*” (*p* < 0.05).

**Table 5 molecules-29-05548-t005:** Phenolic acid concentration in the acceptor fluid during the 24 h permeation study after the application of HKB to the skin.

Time (h)	Gallic Acid	Protocatechuic Acid	*p*-Hydroxybenzoic Acid	*m*-Hydroxybenzoic Acid	Vanillic Acid	Gentistic Acid	Hypogallic Acid
	(µg·cm^−2^)						
1	5.59 ± 0.52	n.d.	n.d.	n.d.	n.d.	n.d.	n.d.
2	7.97 ± 0.87	8.01 ± 1.68	n.d.	n.d.	n.d.	n.d.	n.d.
3	12.31 ± 0.56	9.57 ± 3.06	n.d.	n.d.	n.d.	n.d.	n.d.
5	24.35 ± 1.36	23.38 ± 2.27	n.d.	n.d.	n.d.	n.d.	n.d.
8	51.70 ± 3.44	40.96 ± 2.67	4.27 ± 1.27	7.19 ± 0.36	8.95 ± 1.00	n.d.	n.d.
24	249.73 ± 13.69	97.55 ± 5.31	9.96 ± 2.31	4.13 ± 0.56	15.41 ± 0.55	7.18 ± 0.28	n.d.

n.d.—not identified in the acceptor fluid. The results, obtained from three experiments, are presented as mean values ± standard deviation (±SD).

## Data Availability

The data presented in this study are available in this article.

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
