# Peer review of "Antiaging Properties of Kalanchoe blossfeldiana Ethanol Extract—Ex Vivo and In Vitro Studies"

_molecules, 2024, doi:10.3390/molecules29235548_

Round 1

Reviewer 1 Report

Comments and Suggestions for Authors

Recommendation: Publish after major revisions noted.

Comments:

This manuscript from Justyna Stefanowicz-Hajduk et. al assessed the antioxidant properties and skin permeation of a topical hydrogel containing K. blossfeldiana ethanol extract. The hydrogel containing K. blossfeldiana extract has the potential to be an interesting and new alternative to dermatologic and cosmetic preparations. However, there are some questions while I read the current manuscript. Thus I recommended a major revision is required before this work has been published on Molecules.

(1) In this study, the authors found that the hydrogel containing K. blossfeldiana extract can be considered as an interesting and new alternative to dermatologic and cosmetic preparations. Easy storage conditions are importance for clinical and commercial applications. Can the hydrogel be left in the air for a long time?

(2) The authors had listed a table of compounds categories detected in the ethanol extract of K. blossfeldiana by LC-QTOF-MS. Can the authors Can the author compare the advantages and disadvantages of this extraction method with other reported methods?

(3) Since hydrogels have potential value in future skin wound treatment applications, cytotoxicity experiments are needed to prove their biocompatibility.

(4) All tables and figures should be reorganized for readability and professionalism. 

Comments on the Quality of English Language

Minor editing of English language required.

Author Response

Dear Editor,

We would like to thank Reviewers for critical reading this manuscript and valuable suggestions. We have carefully considered all of the suggestions and made the appropriate corrections and additions (marked in blue).

Reviewer 1

Comment 1: In this study, the authors found that the hydrogel containing K. blossfeldiana extract can be considered as an interesting and new alternative to dermatologic and cosmetic preparations. Easy storage conditions are importance for clinical and commercial applications. Can the hydrogel be left in the air for a long time?

Response: In our studies, due to the fact that no preservative was used, the hydrogel placed in an airless packaging was stored at a temperature of +6 degrees C. However, in order for hydrogels to be used on an industrial scale, they must be preserved, and the selection of the preservative should be tailored to the manufacturer's assumptions, e.g. regarding the shelf life or the requirements for the ecological certification of the cosmetic. Generally, each cosmetic should be protected against exposure to physical factors, e.g. air or light.

Comment 2: The authors had listed a table of compounds categories detected in the ethanol extract of K. blossfeldiana by LC-QTOF-MS. Can the author compare the advantages and disadvantages of this extraction method with other reported methods?

Response: Our research group analyzed different versions of extraction solutions including ethanol, acetone, water or n-butanol. From our experience the highest amount of polyphenolic compounds are detected in acetone and ethanol. For these extraction solutions antioxidant tests and inhibition of elastase indicated similar IC50 values, although inhibition of hyaluronidase is more effective in case of ethanol extract. In addition, for cosmetic formulations ethanol extracts are preferred as more save for skin than acetone or butanol (10.3390/pharmaceutics15051542).

Comment 3: Since hydrogels have potential value in future skin wound treatment applications, cytotoxicity experiments are needed to prove their biocompatibility.

Response: The experiments estimated the cytotoxicity of hydrogel with K. blossfeldiana extract were performed and results are added to the manuscript. We used human fibroblasts HFF-1.  

Comment 4: All tables and figures should be reorganized for readability and professionalism.

Response: The tables and figures were performed according to Molecules requirements. We also added two additional figures to improve our manuscript.

Thank you once again for your time.

Best regards,

Justyna Stefanowicz-Hajduk

Reviewer 2 Report

Comments and Suggestions for Authors

In this manuscript entitled “Antiaging properties of Kalanchoe blossfeldiana ethanol extract – ex vivo and in vitro studies”, the authors report on the investigation of the phytochemical profiles of the ethanol extract of K. blossfeldiana as well as its antioxidant, antielastase and antihyaluronidase activity. In addition, the authors investigated the permeation of selected phenolic acids from a hydrogel containing an extract of K. blossfeldiana through human skin and their accumulation in the skin.

The manuscript is well structured and the conclusions seem promising. Only a few points should be improved and errors corrected:

1) p.2, Introduction section: This section is missing a part that refers to the enzymes elastase and hyaluronidase. For the sake of clarity, the enzymes and the importance of their inhibition in preventing skin aging should be mentioned (up-to-date references should also be added).

2) p.4, Table 1: I suggest that in the last part of the table “Total amount of detected compounds” or simply “Total" be written instead of “Total identified compounds” (since the unidentified compounds are also listed in the table).

3) p. 8-9., lines 186-188. It would be clearer to say: It was found that gallic acid (107.58 ± 15.27 μg×g skin-1) and then protocatechuic acid (72.28 ± 8.08 μg×g skin-1) accumulated the most in the skin.

4) p.17, References section: The pages of the article are missing in the first reference and should be added.

After the above corrections and improvements, the manuscript is suitable for publication in Molecules.

Comments on the Quality of English Language

The text is generally correct, but not ideal for the reader's comfort. Grammatical corrections are necessary.

Author Response

Dear Editor,

We would like to thank Reviewers for critical reading this manuscript and valuable suggestions. We have carefully considered all of the suggestions and made the appropriate corrections and additions (marked in blue).

Reviewer 2

Comment 1: Introduction section: This section is missing a part that refers to the enzymes elastase and hyaluronidase. For the sake of clarity, the enzymes and the importance of their inhibition in preventing skin aging should be mentioned (up-to-date references should also be added).

Response: Information about elastase and hyaluronidase are included in the Introduction section (lines 80-88).

Comment 2: Table 1: I suggest that in the last part of the table “Total amount of detected compounds” or simply “Total" be written instead of “Total identified compounds” (since the unidentified compounds are also listed in the table).

Response: We have corrected this in the table.

Comment 3: p. 8-9., lines 186-188. It would be clearer to say: It was found that gallic acid (107.58 ± 15.27 μg×g skin-1) and then protocatechuic acid (72.28 ± 8.08 μg×g skin-1) accumulated the most in the skin.

Response: Thank you for this comment. We have corrected this sentence.

Comment 4: p.17, References section: The pages of the article are missing in the first reference and should be added.

Response: Thank you. We have added the pages of this article, according to the Molecules requirements.

Thank you once again for your time.

Best regards,

Justyna Stefanowicz-Hajduk

Reviewer 3 Report

Comments and Suggestions for Authors

The article titled “Antiaging properties of Kalanchoe blossfeldiana ethanol extract  – ex vivo and in vitro studies " explores how Kalanchoe blossfeldiana, a plant traditionally used in medicine, can help fight aging and act as an antioxidant. The authors report how this plant extract can be used in skincare products, making it valuable for research on plant-based formulations and skincare. The introduction is well laid out and covers enough material to introduce readers to the subject. The purpose of the publication is briefly and accurately formulated. In the discussion section, the authors discuss their results and compare them with those from the literature, highlighting the benefits of their research results. The article is well written and thoughtful. The experiments were planned, performed and reported correctly. I have no objections to the experimental work.

Despite my positive assessment of the proposed manuscript, I have several questions, the answers to which should be included in the article to make it even more useful for readers.

Can you elaborate on the specific mechanisms by which K. blossfeldiana extract inhibits elastase and hyaluronidase? Are there particular compounds within the extract that are primarily responsible for these effects?

How does the antioxidant activity of K. blossfeldiana compare to other well-studied plant extracts used in cosmetic formulations? Are there specific advantages or disadvantages to using this extract over others?

The manuscript identifies a range of phenolic acids in the extract. How do variations in concentration affect the skin penetration and accumulation of these compounds? Would a higher concentration yield better results?

The study utilized three different antioxidant assays (DPPH, ABTS, and FRAP). How do you account for the differences in IC50 values obtained across these assays? What does this variability suggest about the antioxidant profile of the extract?

If you want your antioxidant activity results to be reliable, why not perform a test that recreates real radicals found in the human body (like superoxide radical, hydroxyl radicals, nitric oxide, peroxyl radicals, etc. )?

Adding your answer in the manuscript would additionally strengthen your article!

Based on the written above I would like to suggest minor revision to the manuscript.

Best regards,  

Author Response

Dear Editor,

We would like to thank Reviewers for critical reading this manuscript and valuable suggestions. We have carefully considered all of the suggestions and made the appropriate corrections and additions (marked in blue).

Reviewer 3

Comment 1: Can you elaborate on the specific mechanisms by which K. blossfeldiana extract inhibits elastase and hyaluronidase? Are there particular compounds within the extract that are primarily responsible for these effects?

Response: In this manuscript, studies were conducted on the ethanol extract of K. blossfeldiana. In the case of extracts containing many different chemical compounds, studies related to the mechanism of enzymatic reactions are not conducted. Extracts are considered as a mixed mechanism, because the molecules present in the extract can affect both the enzyme itself and the enzyme-substrate complex. In the mixture of compounds it is impossible to estimate which of the compounds has a stronger inhibitory effect on enzyme activity: the one in the largest quantity or the one with a higher affinity to the active site but in a lower concentration. When analyzing the effect of extracts on a given enzyme, it is determined whether the extract has an inhibitory effect on the activity of the enzyme and if, its IC50 (the concentration of the extract that inhibits the enzyme by 50%) is determined. In our study, the highest concentration of the analyzed compounds was shown for gallic acid. Plant extracts, in which it is one of the main components, are effective inhibitors of both elastase and hyaluronidase. The examples of manuscripts below indicate this effect:

doi.org/10.1016/j.fitote.2015.01.005 and doi.org/10.1038/s41598-021-95605-3.

This answer has been added to the manuscript.

Comment 2: How does the antioxidant activity of K. blossfeldiana compare to other well-studied plant extracts used in cosmetic formulations? Are there specific advantages or disadvantages to using this extract over others?

Response: K. blossfeldiana is an easy plant to grow even in home conditions. It reproduces and grows relatively quickly. The authors' experience shows that the efficiency of the extraction process is low, due to the small leaves, compared to e.g. K. pinnata or K. daigremontiana. Kalanchoe pinnata, which is the best-studied species of the Kalanchoe genus, is used in cosmetic formulations for a rather anti-inflammatory skin effect. The ethanol extract of K. blossfeldiana, among the three most popular species of Kalanchoe, has the highest antioxidant potential to limit external stress affecting the skin and the highest ability to inhibit the activity of enzymes responsible for the decomposition of skin macromolecules: elastase and hyaluronidase.

Comment 3: The manuscript identifies a range of phenolic acids in the extract. How do variations in concentration affect the skin penetration and accumulation of these compounds? Would a higher concentration yield better results?

Response: The changes in concentration of active compounds can have the significant impact on their after penetration. Generally, the most frequently observed trend is that the higher the concentration of phenolic acids in extracts, the greater the penetration into the skin and through it, which we observed in our study. Gallic acid was present in the greatest amount in the extract and later, after 24 hours, in the acceptor fluid. Sometimes, however, in studies of the permeation of plant extracts, it can be observed that despite the high concentration of some active substances, they permeate to a lesser extent. This may be due to the synergism of many active compounds that can cause an increase or decrease in the permeation of other secondary metabolites.

This part is in the Discussion section.

Comment 4: The study utilized three different antioxidant assays (DPPH, ABTS, and FRAP). How do you account for the differences in IC50 values obtained across these assays? What does this variability suggest about the antioxidant profile of the extract?

Response: Antioxidant experiments included two commonly used tests simulating the neutralization of free radicals in the body: DPPH* and ABTS*, and one test simulating the reducing effect: Ferric Reduction Power (FRAP). In all three tests, very high antioxidant capacities of the ethanol extract of K. blossfeldiana were demonstrated. The differences can occur in such study, however the results are always compared with some standard compound. In the DPPH and ABTS tests, the IC50 of extract was twice as strong as the known antioxidant, ascorbic acid, while the reducing capacities shown in the example of iron reduction were only twice as low as the standard. These values were twice different than the values for ascorbic acid, not ten or hundreds, what indicates that the effect of the extract is significant and can be used to combat free radicals generated in the skin as a result of external and internal factors. Thanks to this, it is able to effectively protect skin macromolecules from broadly understood skin aging.

Comment 5: If you want your antioxidant activity results to be reliable, why not perform a test that recreates real radicals found in the human body (like superoxide radical, hydroxyl radicals, nitric oxide, peroxyl radicals, etc. )?

Response: Antioxidant tests conducted in this manuscript are standard tests used for preliminary antioxidant evaluation of the extract in vitro. The K. blossfeldiana extract showed high, satisfactory antioxidant results in DPPH, ABTS and FRAP tests and we agree that further studies in vivo should be performed. Our results will be the basis for undertaking this type of research in the future. This statement has been added in the Conclusion.

Thank you once again for your time.

Best regards,

Justyna Stefanowicz-Hajduk

Round 2

Reviewer 1 Report

Comments and Suggestions for Authors

Accept in present form

Comments on the Quality of English Language

 Minor editing of English language required.

Author Response

Dear Editor,

we would like to thank the Reviewer for valuable comments and time.

Best regards,

Justyna Stefanowicz-Hajduk

Department of Biology and Pharmaceutical Botany

Medical University of Gdańsk